# Simulation Study on Anti-Interference Performance Degradation of GIS UHF Sensors Based on Substation White Noise Reconstruction

**DOI:** 10.3390/s26010303

**Published:** 2026-01-02

**Authors:** Lujia Wang, Yongze Yang, Zixi Zhu, Haitao Yang, Jie Wu, Xingwang Wu, Yiming Xie

**Affiliations:** 1School of Electrical Engineering, China University of Mining and Technology, Xuzhou 221116, China; yongze.yang@cumt.edu.cn (Y.Y.);; 2Electric Power Research Institute, State Grid Anhui Electric Power Co., Ltd., Hefei 230601, China

**Keywords:** partial discharge, sensors, GIS, white noise

## Abstract

The ultra-high frequency (UHF)-based partial discharge (PD) detection technology for gas-insulated switchgear (GIS) has achieved large-scale applications due to its high sensitivity and real-time monitoring capabilities. However, long-term service-induced antenna corrosion in UHF sensors may lead to degraded reception characteristics. To ensure the credibility of monitoring data, on-site sensor calibration under ambient noise conditions is required. This study first analyzes the time–frequency domain characteristics of white noise received by UHF sensors in GIS environments. Leveraging the transceiver reciprocity principle of sensors, a noise reconstruction method based on external sensors is proposed to simulate on-site white noise. Subsequently, CST simulation models are established for both standard and degraded sensors, quantifying the impact of factors like antenna corrosion on performance parameters such as echo impedance S11 and voltage standing wave ratio (VSWR). Finally, the two sensor models are coupled into GIS handholes for comparative simulation analysis. Results show that antenna corrosion causes resonant frequency shifts in sensors, reducing PD signal power by 55.27% and increasing noise power by 64.11%. The signal-to-noise ratio (SNR) decreases from −9.70 dB to −15.34 dB, with evident waveform distortion in the double-exponential PD pulses. These conclusions provide theoretical references for on-site UHF sensor calibration in noisy environments.

## 1. Introduction

Gas-insulated switchgear (GIS), as a critical component of power systems, directly impacts grid security through its insulation reliability [1,2]. Partial discharge (PD), serving as both an early indicator of insulation degradation and a primary accelerator of insulation failure [3,4,5], has made ultra-high frequency (UHF)-based PD detection technology a mainstream online monitoring method due to its high sensitivity (up to 10 pC level) and broadband response (0.3–3 GHz) [6,7,8,9]. However, long-term service-induced dielectric aging and metal corrosion may shift antenna resonance points of UHF sensors, potentially masking PD signals. Furthermore, complex electromagnetic environments (e.g., Gaussian white noise) in substations distort UHF sensor signals [10,11,12], exacerbating calibration challenges and undermining reliability.

Existing studies predominantly focus on noise-free sensor performance analysis, such as GIS-embedded UHF sensor simulations [13] and PD localization verification [14]. Systematic research on sensor anti-interference degradation under white noise–PD signal coupling remains scarce. Addressing this gap, this paper proposes a CST simulation-based methodology to investigate UHF sensor performance degradation in white noise environments. By establishing standard/degraded sensor models, we quantify the impacts of antenna corrosion and dielectric loss on voltage standing wave ratio (VSWR) and signal amplitude. Combined with white noise reconstruction injection mechanisms, this enables quantitative evaluation of signal-to-noise ratio (SNR) and amplitude stability within 0.5–3 GHz. The findings provide theoretical support for diagnosing sensor anti-interference degradation in GIS condition monitoring.

## 2. UHF White Noise Characteristics Analysis and Reconstruction

UHF sensors in GIS PD detection are susceptible to electromagnetic noise. This chapter systematically analyzes white noise generation mechanisms and statistical properties, proposing a MATLAB R2025a-based time-domain waveform reconstruction method to optimize noise suppression algorithms.

### 2.1. White Noise Characteristics and Simulation Reconstruction

Substation white noise exhibits full-spectrum characteristics, making conventional filtering ineffective. During on-site sensor calibration, such noise obscures PD pulses, causing timing misjudgment and localization errors [15,16,17,18,19].

Due to temperature variations and resistance differences in field equipment, noise cannot be represented by a single formula. Thermal noise follows a Gaussian distribution, where variance reflects noise energy. This study calculates background noise variance *σ*^2^ from PD-free measurements and sets white noise parameters accordingly. For a white noise signal x(t), the variance is computed as follows:(1)$$\sigma^{2}=\frac{1}{N-1}\sum_{i=1}^{N}{\left(x_{i}-\mu \right)}^{2}$$
where *N* is the sampling count, *x_i_* is the *i*-th sample, and *μ* is the mean (ideally *μ* ≈ 0).

As shown in Figure 1, field measurements at a 550 kV GIS substation yielded σ2 = 1.82 V^2^, corresponding to a simulation amplitude coefficient Awhite = 1.35 V. 3σ ≈ 4 V peak ensures compliance with field noise characteristics.

MATLAB-based Gaussian white noise sequences are generated using the randn function. Time-domain verification (Figure 2a) shows random oscillations within [−4.5 V, 4.5 V], while frequency-domain analysis (Figure 2b) confirms flatness (0–1 GHz amplitude: 0–0.4 V, power spectral density fluctuation <±2 dB). Kolmogorov–Smirnov tests show >95% consistency with measured data:(2)$$X[n]=\sum_{k=0}^{N-1}x[k]e^{-j\frac{2\pi }{N}nk}$$

The results are shown in Figure 2b, with amplitude fluctuations in the range of 0–0.4 V in the 0–1 GHz band and power spectral density fluctuations of less than ±2 dB, which are in accordance with the assumption of white noise spectral flatness. The signal time accuracy reaches 0.1 ns, which can be directly injected into the UHF sensor for noise signal reconstruction simulation.

The simulation model is verified by the third-order statistics: The kurtosis value, which measures the ‘tailedness’ of the probability distribution, was found to be κ = 2.98 ± 0.15, close to the theoretical value of 3 for a Gaussian distribution and the peak sidelobe ratio of the autocorrelation function is lower than −30 dB, which indicates that the noise sequence possesses a good Gaussianity and irrelevance, and it can be used as a typical Gaussian electromagnetic white noise.

### 2.2. Equivalence Validation of Reconstructed Signals

The reconstructed Gaussian white noise is imported into the CST software (CST2025) as an excitation signal. When injected into a UHF sensor model, the output noise waveform (Figure 3a) retains Gaussian characteristics.

Fast Fourier Transform (FFT) analysis (Figure 3b) confirms white noise distribution in the 0–1 GHz band.

## 3. UHF Sensor Modeling and Performance Degradation Analysis

In this section, the performance degradation simulation of the sensor is performed based on the possible damage or performance degradation of the sensor in reality. The antenna arm inside the sensor is made of metal, such as copper or iron. Therefore, the antenna radiating unit, i.e., the metal arm of the antenna, may be oxidized and corroded or even broken because of the damage to the airtight cover and other reasons, especially in the coastal environment with high humidity, which leads to the accelerated oxidation of the antenna arm. Therefore, for the possible antenna rust fracture failure, this section changes the antenna input impedance by changing the number of antenna coils N, antenna width Tw, antenna connection, and other parameters to realize the simulation of the impact of changing the resonance point on the reception characteristics of the sensor.

### 3.1. Standard Ultra-High-Frequency Sensor Simulation

Establish a simulation model of the antenna radiation unit within the CST simulation software. Set the antenna trace width Tw to 6.7 mm, the number of turns to 3.7 turns, the antenna inner diameter r_0_ to 5 mm, and the antenna trace height Th to 0.035 mm. For arm 1, the antenna start angle is 0°, and the antenna end angle is 0°; for arm 2, the antenna start angle is 180°, and the antenna end angle is 0°. The substrate thickness is set to 1.2 mm, with dimensions of 120 mm in length and width. The material is specified as FR-4 epoxy resin dielectric.

This section identifies the optimal structural parameters for the Archimedean spiral antenna radiating element within the 700–3000 MHz operating frequency range by comparing the antenna’s S11 parameters and VSWR parameters, specifically evaluating its return loss and reception loss.

As shown in Figure 4, this model simulates the antenna radiation unit of an ultra-high-frequency partial discharge sensor by etching two Archimedean spiral-shaped metallic conductors with PerfectE boundary conditions onto a 1.2 mm-thick epoxy resin dielectric substrate.

The radiating element employs a lumped port excitation, with the central rectangular region serving as the antenna excitation source, as illustrated in Figure 5. Simulations were conducted within the 500 MHz to 3000 MHz frequency band to calculate the antenna’s S11 parameters and voltage standing wave ratio (VSWR).

Individual simulation of the radiation element yielded the S11 parameter results shown in Figure 6. As illustrated, within the frequency range of 0.7 GHz to 3 GHz, the radiation element’s S11 parameter remains stable at approximately −10 dB. This wide bandwidth and consistent gain performance largely meet the requirements for ultra-high-frequency partial discharge sensor antennas.

### 3.2. Simulation of Oxidation and Corrosion on Antenna Rotating Arms

The metal arm of the antenna, after air corrosion, causes the surface layer of the metal arm to turn into metal oxide and lose the signal radiation ability, so the surface layer of oxide can be regarded as the metal arm becoming thinner and shorter. It is important to note that this modeling approach primarily captures the geometric effects of corrosion (i.e., the effective thinning and reduction in the antenna conductor) on resonance characteristics and impedance matching. It does not simulate the complex electrochemical processes or frequency-dependent material property changes associated with the full physics of corrosion. Under mild corrosion conditions, the antenna line width Tw is reduced to 5 mm, and the number of antenna coils N is reduced to three turns, as shown in Figure 7a above. When the antenna metal arm is severely corroded, only the uncorroded area inside the arm remains in the actual radiating area, and a too-thin antenna width may cause the antenna arm to break and the number of actual radiating circles to be greatly reduced. Under the condition of heavy corrosion, the antenna line width Tw is reduced to 2 mm, and the number of antenna coils N is reduced to 1.5 turns.

The S11 parameter is the size of return loss caused by impedance incompatibility, and the VSWR parameter is the voltage standing wave ratio, which can measure the degree of impedance matching between the antenna and transmission line. In this section, the above two parameters are chosen to react to the extent that the antenna reception characteristics are reduced with corrosion.

From Figure 8 and Table 1, the effects of corrosion on antenna performance are evident. After mild corrosion, the antenna performance is slightly degraded: the average S11 parameter changes from −13.85 dB (standard) to −14.23 dB, while the VSWR increases from 1.94 to 2.19, indicating a mild impedance mismatch. In contrast, severe corrosion leads to significant performance degradation. The average S11 parameter rises to −9.07 dB, and the VSWR increases dramatically to 4.18, demonstrating a severe impedance mismatch. Across the frequency range of 500 MHz to 1.5 GHz, the performance degradation is particularly pronounced, with parameters falling considerably below those of the standard antenna model.

These changes indicate that corrosion causes impedance mismatch between the antenna input impedance and the feeding balun, which reduces the antenna return loss. Consequently, the signals emitted by the antenna’s radiating elements are affected, leading to decreased voltage amplitude and diminished sensitivity. The degradation is more severe in the low-frequency band (500 MHz to 1 GHz) for both corrosion levels, though the effect is substantially more pronounced under severe corrosion conditions.

Figure 9 presents the 2D radiation gain patterns for lightly and heavily corroded antennas at a frequency of f = 1.5 GHz. 1.5 GHz is a representative center frequency within the sensor’s operating band (0.7–3 GHz) and a common frequency point for partial discharge signal energy. The diagram reveals that corrosion degrades the antenna’s directional characteristics, resulting in non-directional gain perpendicular to the antenna’s radiation elements and an asymmetrical radiation pattern. The maximum gain is 3.35 dBi, representing 61% of the standard antenna model’s gain. Under these conditions, the antenna’s capability for detecting partial discharges is significantly diminished.

## 4. Simulation and Calibration of Sensor Performance in White Noise Environment

In order to realize the sensor performance calibration under a white noise environment, this section simulates the process using three UHF sensors [20,21]. Two UHF sensors are installed at the handhole of the 500 kV GIS, as shown in Figure 10. The No. 1 sensor is used as a local discharge signal-emitting sensor, and the No. 2 sensor is a sensor to be calibrated, which receives the signal of the local discharge as well as the white noise in the environment. The No. 3 sensor is placed in the insulator of the GIS basin, which is an insulating material, so that the electromagnetic wave from outside can enter the interior of the GIS through this place, and interfere with the built-in UHF sensor to receive the signal. The built-in UHF sensor receives local emission signals. Detailed GIS dimensions are shown in Table 2.

A double-exponential decay model is used for the local discharge pulse signal:(3)$$I(t)=A(e^{-\alpha (t-t_{0})}-e^{-\beta (t-t_{0})})$$

As shown in Figure 11, the simulated local-amplitude pulse signal shows a rapid rise in the pulse to a peak value at 20 ns, followed by an exponential decay. The peak value of the local-amplified signal is 2.61 V. The signal time-domain waveform exhibits a typical double-exponential feature: a fast rise (about 2 ns) and a slow fall (about 50 ns).

Sensors with different degradation degrees are placed at the position of sensor No. 2 to be tested, and the difference in the received signal waveforms is simulated and analyzed. Figure 12 shows the received signal of the probe placed at the No. 2 sensor, which has not been converted by the electromagnetic signal of the receiving antenna, so it is a standard received waveform. Comparison of the received signal with the sensor in the figure below shows the degree of distortion of the sensor’s received signal. As can be seen from Figure 13, with the deepening of the degree of corrosion, the received signal waveform is gradually distorted.

Quantitative processing of sensor-received signals was performed to calculate the power of partial discharge signals and noise signals at different degradation levels. In signal processing, when direct measurement of actual power (watts, W) is impractical, equivalent power is often calculated via voltage amplitude. The signal power formula is as follows(4)$$P=\frac{1}{N}\sum_{n=1}^{N}V{\left(n\right)}^{2}$$
where *V*(*n*) denotes the mean-subtracted voltage signal, and *N* represents the signal length.

The signal-to-noise ratio of the received signal under noise interference was determined by taking the logarithm of the ratio of the partial discharge signal power *P*_signal_ to the noise power *P*_noise_. The signal-to-noise ratio is calculated as follows:(5)$$\mathrm{SNR}(\mathrm{dB})=10\log_{10}\left(\frac{P_{\mathrm{signal}}}{P_{\mathrm{noisc}}}\right)$$

The signal-to-noise ratios were calculated for three different degradation degree sensors. From Figure 13 and Table 3, it can be seen that that due to material aging, the partial discharge sensor’s resonance point shifted. This shift reduced the local discharge signal band receiving ability and the received noise amplitude, which ultimately resulted in an increase in the received signal’s signal-to-noise ratio.

## 5. Conclusions

In this paper, for the problem of degradation calibration of GIS UHF local-amplitude sensor performance in a white noise environment, a research method of degradation of sensor reception characteristics based on noise reconstruction and electromagnetic field simulation is proposed, and the main conclusions are as follows:The Gaussian white noise simulation model constructed by measured noise ANOVA, its craggy value and autocorrelation characteristics are highly consistent with the measured noise, and it can reproduce the white noise interference in the 0–1 GHz band in CST.Antenna corrosion causes the VSWR to deteriorate from 1.94 (normal) to 4.18 (severe corrosion), the S11 parameter deterioration reaches 4.78 dB, and the gain of the directional map is attenuated by 39%. The mapping relationship of “corrosion line width—resonance point offset—receiver characteristic degradation” is established.In the superposition of white noise and local amplifier pulse, the power of the local amplifier signal received by the sensor decreases by 55.27%, the power of the noise signal increases by 64.11%, and the signal-to-noise ratio decreases from −9.70 dB to −15.34 dB. The simulation results prove that the aging of the sensor material greatly reduces the signal-to-noise ratio of the sensor signal and provides theoretical references for the on-site calibration of the degraded sensors.

## Figures and Tables

**Figure 1 sensors-26-00303-f001:**
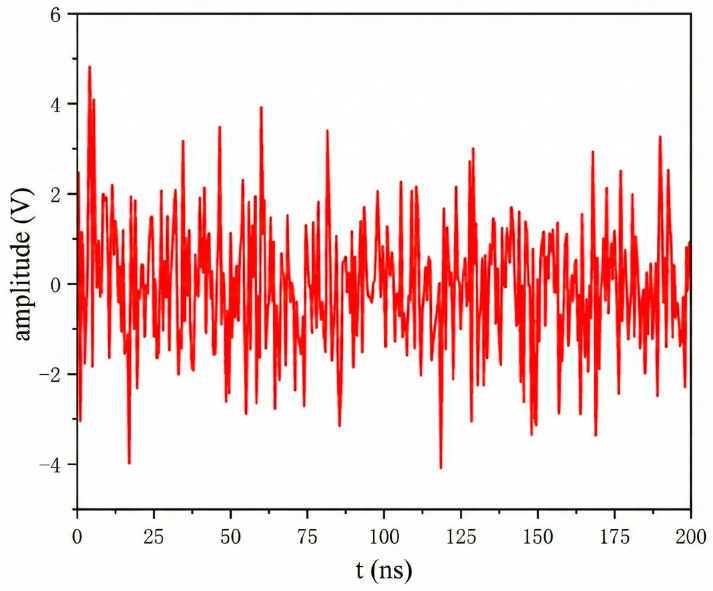
Measured background noise at a 550 kV GIS substation.

**Figure 2 sensors-26-00303-f002:**
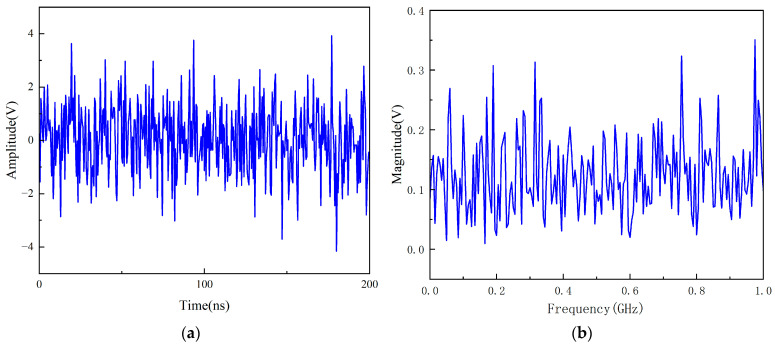
(**a**) MATLAB Gaussian white noise time domain waveforms; (**b**) MATLAB Gaussian white noise frequency domain waveforms.

**Figure 3 sensors-26-00303-f003:**
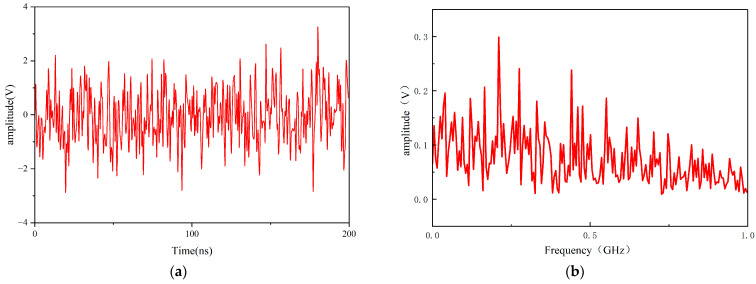
(**a**) Sensor output noise waveform; (**b**) FFT of noise waveforms.

**Figure 4 sensors-26-00303-f004:**
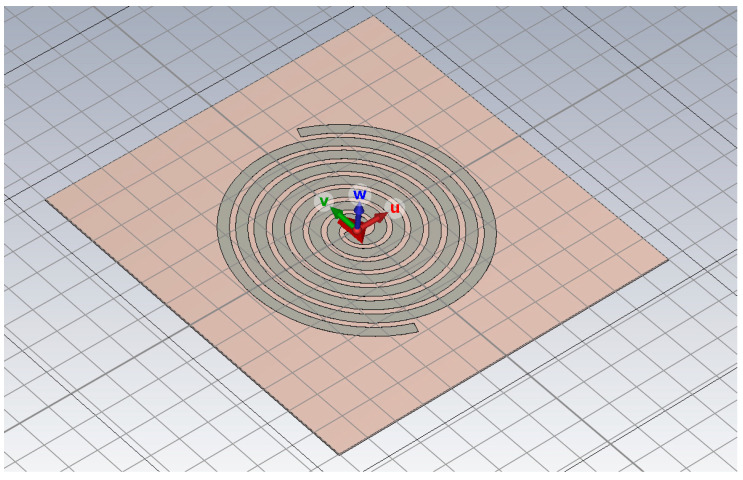
Archimedean spiral antenna radiating element.

**Figure 5 sensors-26-00303-f005:**
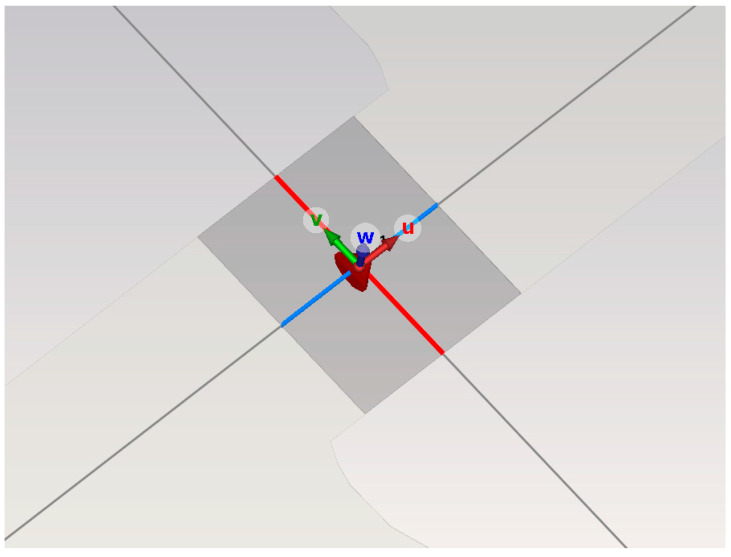
Excitation source of the Archimedean spiral antenna radiation unit.

**Figure 6 sensors-26-00303-f006:**
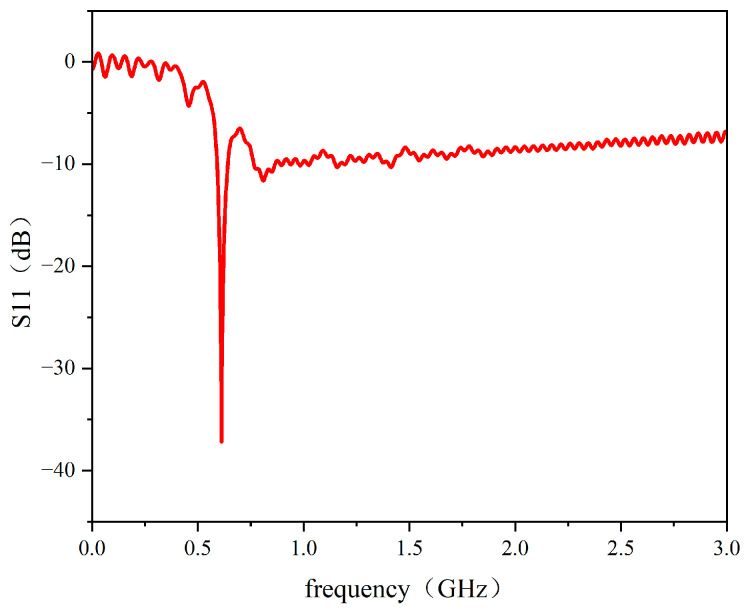
Radiating alement S11 parameter.

**Figure 7 sensors-26-00303-f007:**
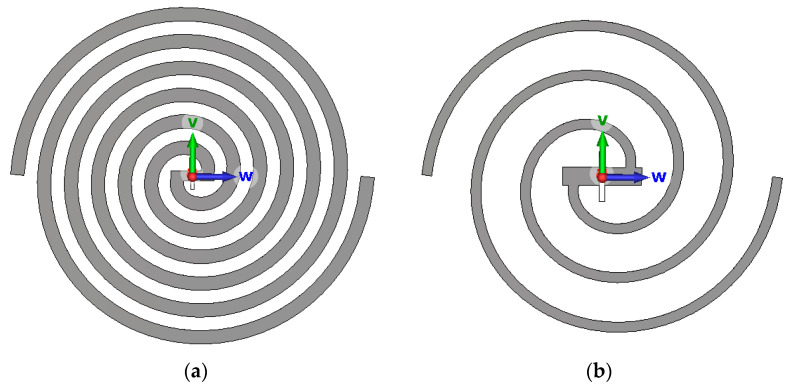
(**a**) Mildly rusted antenna; (**b**) heavily rusted antenna.

**Figure 8 sensors-26-00303-f008:**
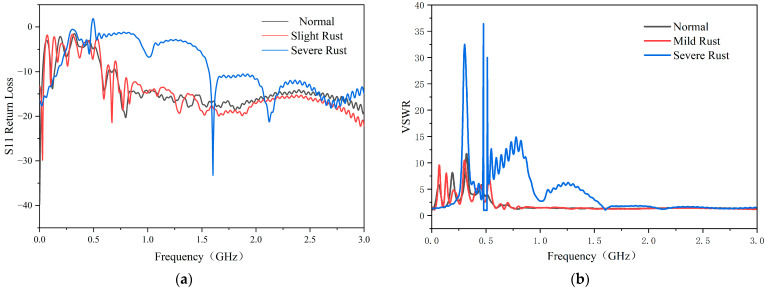
(**a**) Comparison of S11 parameters for standard, mildly corroded, and severely corroded antennas, and (**b**) comparison of VSWR parameters for standard, mildly corroded, and severely corroded antennas.

**Figure 9 sensors-26-00303-f009:**
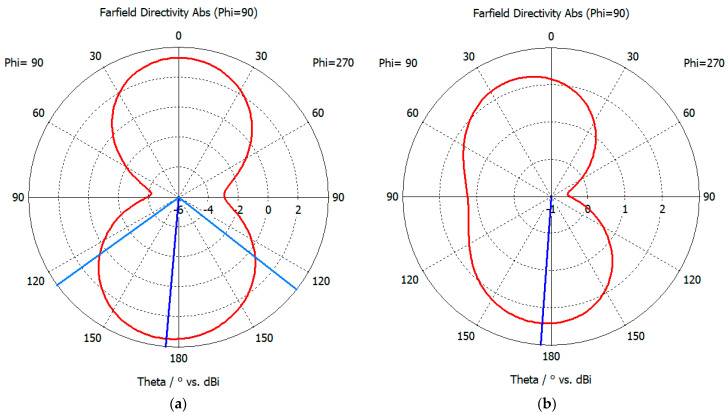
(**a**) Radiation patterns of mild corrosion; (**b**) radiation patterns of severe corrosion.

**Figure 10 sensors-26-00303-f010:**
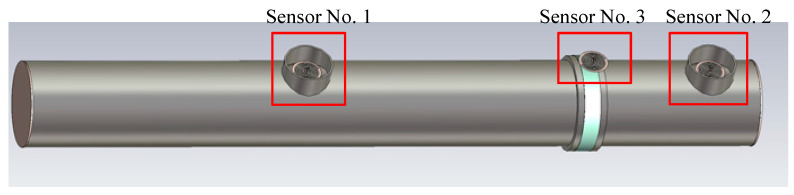
Simulation diagram of the built-in sensor placed at the GIS hand hole.

**Figure 11 sensors-26-00303-f011:**
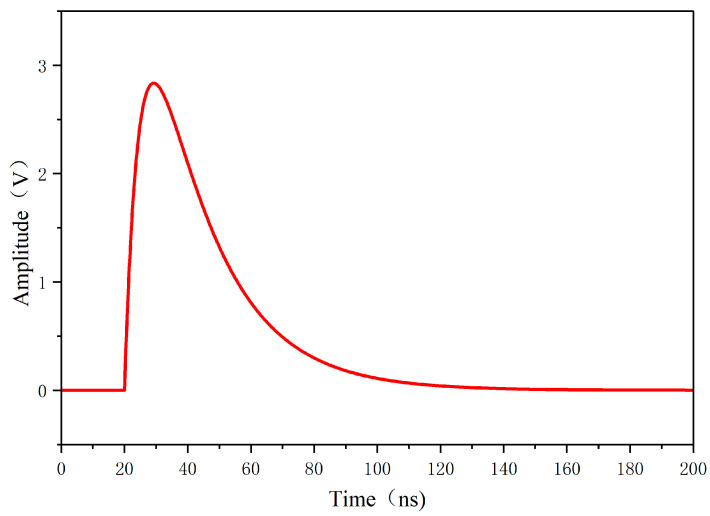
Analog local discharge pulse signal.

**Figure 12 sensors-26-00303-f012:**
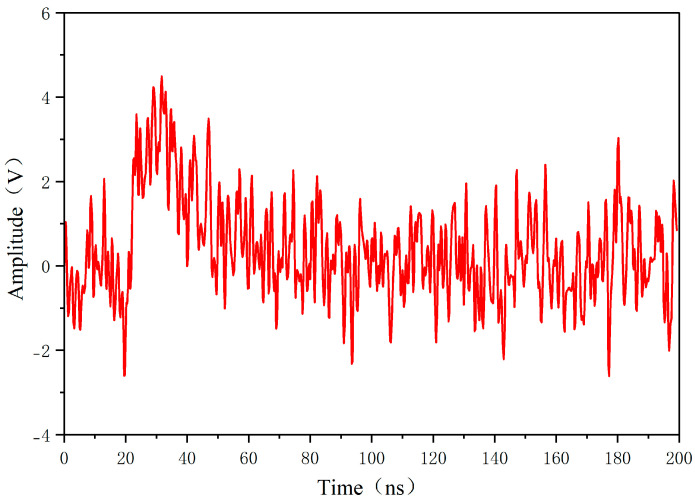
Signal received by the probe at sensor No. 2.

**Figure 13 sensors-26-00303-f013:**
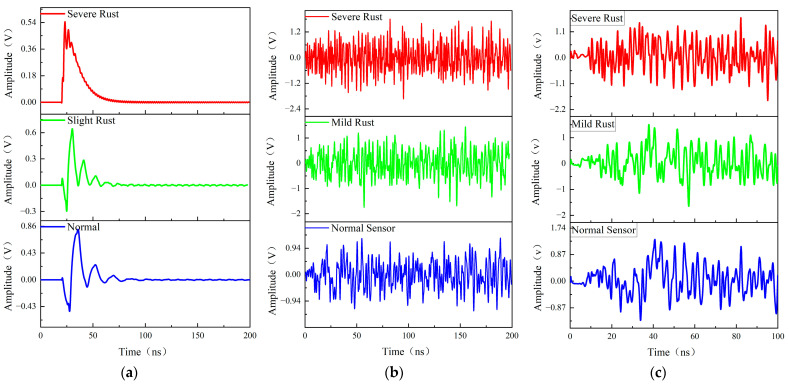
Comparison of sensor signal waveforms at different degradation levels. (**a**) Partial discharge signal excitation, (**b**) noise signal excitation, and (**c**) partial discharge signal mixed with noise signal excitation.

**Table 1 sensors-26-00303-t001:** Comparison of sensor performance degradation.

Sensor Degradation	VSWR	S11 (Return Loss)
Normal	1.94	−13.85 dB
Slight corrosion	2.19	−14.23 dB
Heavy corrosion	4.18	−9.07 dB

**Table 2 sensors-26-00303-t002:** 500 kV GIS simulation modeling dimensions.

Component	Dimensions	
GIS pipe body	4.5 m	
GIS tube radius	Inner wall 0.26 m	Outer wall 0.27 m
Inner conductor	radius 0.25 m	
Handhole radius	Inner wall 0.14 m	Outer wall 0.15 m
Handhole spacing	2.5 m	
Basin insulator	Radius 0.3 m	Width 0.1 m

**Table 3 sensors-26-00303-t003:** Comparison of sensor reception characteristics for different degradation levels.

Sensor Degradation	Signal Power (V^2^)	Noise Signal Power (V^2^)	SNR
Normal	0.0237	0.2207	−9.70 dB
Slight Corrosion	0.0096	0.2541	−14.24 dB
Heavy corrosion	0.0106	0.3622	−15.34 dB

## Data Availability

Data are contained within the article.

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
