# Peer review of "Simulation Study on Anti-Interference Performance Degradation of GIS UHF Sensors Based on Substation White Noise Reconstruction"

_sensors, 2026, doi:10.3390/s26010303_

Round 1

Reviewer 1 Report

Comments and Suggestions for Authors

The authors reconstruct Gaussian white noise from measured statistics and build CST models for standard and corroded UHF sensors to quantify resonance shifts, impedance mismatch, and SNR degradation. The approach combines noise reconstruction with degradation modeling for GIS calibration. The detailed comments are as follows.

1. The noise model is generated only from variance and randn, and validated in 0–1 GHz, while actual substation noise often contains colored components and harmonics. A brief comparison with the measured spectrum would help clarify that the model represents ideal Gaussian noise rather than full field noise.

2. Noise validation is limited to 0–1 GHz, while the paper states the operational band is 0.3–3 GHz. The authors should explain why only the lower band is examined and whether this is due to noise distribution or CST excitation limits.

3. Equation (2.1) mixes the forms 1/N and 1/(N−1). The chosen definition of variance should be stated clearly and kept consistent. Equation (2.2) would benefit from a standard DFT form to avoid ambiguity in scaling and frequency indexing.

4. The term “craggy value” is nonstandard and may confuse readers. It should be replaced with the conventional term (e.g., kurtosis) with a clear definition.

5. The corrosion model alters line width and turn count but does not explain its physical basis. A short note stating that the model captures only geometric effects on resonance—rather than full corrosion physics—would clarify its scope.

6. Table 1 and the related text contain inconsistent descriptions of VSWR changes under corrosion. The authors should align the wording with the numerical values and remove contradictions.

7. Section 3.2 repeats similar statements and uses inconsistent terms (“slight” vs. “significant” degradation) for the same data. These paragraphs should be merged and streamlined.

8. Table 2 lists identical radii for the GIS inner wall and inner conductor, which is physically impossible and should be corrected.

9. Radiation patterns are shown only at 1.5 GHz. A brief explanation of whether this frequency is representative of the band would improve clarity without requiring additional simulations.

Comments on the Quality of English Language

The manuscript is generally understandable, but several sections contain inconsistent terminology, ambiguous wording, and minor grammatical issues that require careful editing for clarity and technical precision.

Author Response

We have revised the manuscript in accordance with your comments. Please see the attached file.

Reviewer 2 Report

Comments and Suggestions for Authors

The authors present the results of simulation studies in the presented paper.
This is an interesting paper, but I see the following problems:
Lines 26, 27 and 267 say:
"The signal-to-noise ratio (SNR) decreases from -9.70 dB to -15.34 dB …" 
So the SNR is clearly negative, i.e. the useful signal is lower than the noise. The authors should explain why this is happening. How can useful information be extracted in a real system with such strong noise? - this is not clear to the reader. With the above in mind, let us analyze the results in Table 3: What are the units in columns "Local amplified" and "signal power"? If the signal and noise power were measured in dB relative to the same reference level, the SNR is positive. How were the values ​​in Table 3 measured?
The values ​​in column "Noise signal power" are the same as the SNR values. Noise signal power is absolutely not the same as SNR. The authors must clarify the above doubts.
Tabel 3 - Sensor degradation: Normal. Not Norma.
Figure 6 - Please write "frequency" GHz in English.

Author Response

(The authors gave the same response as above.)

Round 2

Reviewer 1 Report

Comments and Suggestions for Authors

The manuscript presents a meaningful study and the revisions have improved its clarity. The modeling assumptions, noise reconstruction process, and corrosion analysis are now better explained. A few sections could still benefit from minor polishing to enhance consistency and readability. With careful final editing, the manuscript is suitable for publication.

Comments on the Quality of English Language

The English is generally clear, with only minor wording and consistency issues that can be improved through routine editing.

Reviewer 2 Report

Comments and Suggestions for Authors

I thank the authors for their clarifications. The new version is more understandable to the reader. I accept the new version of the paper.